# Phenolic compounds from *Arthrospira platensis* and *Chlorella vulgaris* enhance growth, digestive function, antioxidant capacity, and immune-related gene expression in Nile Tilapia

Eman Y. Mohammady[1], Mohamed R. Soaudy[2], Mohamed A. Elashry[2], Abeer M. A. Mahmoud[1], Soaad A. Sabae[1], Anisa Mitra[3], Ehab R. El-Haroun[4,5]*, Mohamed S. Hassaan [2]*

1 National Institute of Oceanography and Fisheries, NIOF, Cairo, Egypt, 2 Department of Animal Production, Fish Research Laboratory, Faculty of Agriculture at Moshtohor, Benha University, Benha, Egypt, 3 Department of Zoology, Sundarban Hazi Desarat College, Affiliated to University of Calcutta, Kolkata, West Bengal, India, 4 Department of Animal Production, Fish Research Laboratory, Faculty of Agriculture at Cairo University, Cairo, Egypt, 5 Department of Integrative Agriculture, College of Agriculture and Veterinary Medicine, United Arab Emirates University, Al Ain, Abu Dhabi, United Arab Emirates

* Mohamed.hassaan@fagr.bu.edu.eg (MSH); ehab.reda@uaeu.ac.ae (ERE-H)

## Abstract

Microalgae have many bioactive compounds as well as nutritional properties and have been used as a functional feed supplement to improve the health status of fish and their performance. Thus, the current trial was conducted to test the impacts of dietary supplement of phenolic compounds extracted from either *Arthrospira platensis* (PCA) or *Chlorella vulgaris* (PCC) on Nile tilapia (initial body weight: 1.52±0.10 g) growth performance, digestive and hepatic antioxidant enzymes activity and immune related gene expression over a period of 70 days. Therefore, three isoproteic and isolipidic diets were formulated, and the experimental fish were fed these diets to satiation. The basal diet did not contain any supplements (control diet), whereas the other two diets were supplemented with 50 mg/kg of PCA and PCC, respectively. Compared with the control diet, growth parameters and survival rates were significantly enhanced ($P < 0.05$) by dietary supplementation of phenolic compounds of PCA and PCC. Most digestive enzymes were recorded in fish fed with phenolic extract of *A. platensis*. The ALT and AST values of fish fed diet supplemented with phenolic compounds extract from either PCA or PCC improved significantly compared with the basal diet. In addition, blood profile including serum total protein, albumin and globulin amounts augmented significantly in fish fed a diet enriched with phenolic compounds extracted of PCA and PCC ($P < 0.05$). The maximum immune response parameters such as phagocytic activity, lysozyme and IgM activities were recorded for fish fed diets enriched with phenolic extract of PCA ($P < 0.05$). The activities of antioxidant enzymes were significantly higher ($P < 0.05$) in fish fed diets enriched with phenolic extract of *A. platensis* and *C. vulgaris* compared with the basal diet (control).

**Data availability statement:** All relevant data are within the paper.

**Funding:** The author(s) received no specific funding for this work.

**Competing interests:** No conflict of interest.

Diet containing phenolic extract of PCA and PCC showed (*P< 0.05*) up-regulated transcripts of interferon gamma (*IFN-γ*) and interleukin 1β (*IL-1 β*), but heat shock protein 70 (*HSP-70*) genes were down-regulated. Fish fed phenolic extract of PCA showed the highest levels of *IFN-γ* and *IL-1 β* gene expression. Based on the findings achieved, the supplemental diets containing either PCA or PCC modulated growth performance, blood profile, enzymes activities, immune responses and related immune genes, with PCA providing the most effective response.

## 1. Introduction

A perfect balance between intensification of production systems and fish health is required for sustainable aquaculture. The use of chemicals to control disease outbreaks and infections has led to the emergence of resistant bacterial pathogens and the accumulation of these substances in the environment [1]. Nile tilapia is the second most farmed fish species worldwide. Its production has quadrupled over the past decade, largely due to its suitability for aquaculture and the stability of its market price [2,3]. With the increasing global demand for tilapia [4], farmers have adopted intensive culture techniques for its production. Consequently, the industry has become more susceptible to pathogen infections and disease outbreaks [5]. Among natural phytochemical additives, dietary polyphenols and polyphenol-rich diets have shown great promise in modern aquaculture [6,7]. They can scavenge oxygen and nitrogen-derived free radicals, modulate antioxidant enzyme activity, and influence cell-to-cell signaling, thereby supporting healthy metabolic functions and improving fish performance without causing environmental harm [8–10]. As a result, polyphenols can be considered as a promising alternative to routine compounds traditionally used in aquatic animal farming [11,12]. Polyphenols, a major class of phytochemicals and secondary metabolites produced by plants and microalgae, are known for their antioxidant and pigmenting properties [13]. These compounds, especially catechins like epigallocatechin-3-gallate, have been shown to improve lipid metabolism, reduce lipid oxidation, and boost immune function in mammals [14]. Polyphenols have recently attracted attention as feed additives in aquaculture, which improve fish growth, increase in meat quality, immune and oxidative response [15,16]. Numerous trials have been conducted to validate their role as functional feed supplements and to highlight their positive effects on aquaculture sustainability in various fish species [17–27].

Microalgae *Chlorella vulgaris* and *Arthrospira platensis* are frequently used as nutraceutical supplements and/or added in aquaculture [28]. This metabolite has hepatoprotective, anti-inflammatory, and antioxidant properties as well as redox mechanisms, reducing reactions, and oxygen quenching [29–32]. In this perspective, *C. vulgaris* and *A. platensis* are among the most well-known microalgae because of their high levels of phenolic compounds, volatile compounds, sterols, vitamins, polysaccharides, and pigments [33]. However, *A. platensis* has stronger antioxidant properties than *C. vulgaris* because it contains more phenolic compounds [34]. Recently,

some research in aquaculture has highlighted the importance of including polyphenols in aquafeeds as growth promoters and immunostimulants. The inclusion of polyphenols in canola meal-based diet at 400 mg kg$^{-1}$ level improved the growth performance of common carp in terms of FCR and weight gain. Indeed, a small addition of 0.02% tea polyphenols (TPs) to the diet of coho salmon noticeably boosted their growth [35]. Pham et al. (2006) [36] found that increasing the amount of *Hizikia fusiformis* (a rich source of polyphenols) in the diet led to better growth in olive flounder. Jian and Wu (2004) [37] reported higher weight gain in carp when diets were supplemented with a mixture of Chinese angelica root and astragalus root as sources of polyphenols. Munglue (2014) [38] found growth performance to be significantly better in Nile tilapia fed with 1% *Nelumbo nucifera* (Lotus) peduncle extract (NNPE), rich in polyphenols. Limited data are available on the comparative effects of phenolic compounds from *Arthrospira platensis* and *Chlorella vulgaris* on aquatic animals, especially Nile tilapia are the most widely cultivated of any farmed fish. Therefore, the current trial is the chief effort to examine the impacts of phenolic compounds extracted from either *C. vulgaris* or *A. platensis* at level 50 mg kg$^{-1}$ diet on performance, feed utilization efficacy, serum biochemical parameters, antioxidant activities, immunological response and its related genes expression of Nile tilapia.

## 2. Materials and methods

### 2.1 Extraction of phenolic compound

Extracts of phenolic compounds (Polyphenols and flavonoids) from the *A. platensis (*PPA) and *C. vulgaris* (PPC) biomass samples were detected according to the method described by Li et al. (2007) [17]. After that, the residues were dissolved in petroleum ether (10 mL). The detection of polyphenols antioxidants carried out by HPLC (Agilent, USA), (Table 1 and Fig 1A and 1B). The extract was dissolved in methanol at a concentration of 100 µg/mL and its UV spectrum was determined in the UV region (200–400 nm) and in UV- Vis spectrophotometer (Shimadzu 1800) against methanol blank. The samples were spotted in silica gel plate to obtain TLC chromatogram. The pure compound was evaporated by drying and the precipitated residue dissolved in HPLC grade methanol. The extract was injected into an LC- 8A Shimadzu 72 C18 column with HPLC grade acetonitrile- water gradient system over 15 min at a flow rate of 0.5 mL/min with detection at 254

**Table 1. Bioactive compounds identified from the *A. platensis* and *C. vulgaris*.**

| Name | Amount (ug ml$^{-1}$) | | Retention Time (min) | |
|---|---|---|---|---|
| | *A. platensis* | *C. vulgaris* | *A. platensis* | *C. vulgaris* |
| Gallic acid | – | – | 4.318 | 4.318 |
| Catechol | – | – | 8.226 | 8.226 |
| p- Hydroxy benzoic acid | 7.46184 | 3.119 | 9.241 | 9.291 |
| Caffeine | 6.41878e | 5.30579 | 10.309 | 10.194 |
| Vanillic acid | 2.15797 | 5.97314 | 10.880 | 10.614 |
| Caffeic acid | 3.46271 | 1.2753 | 11.228 | 11.217 |
| Syringic acid | 3.231 | – | 11.690 | 11.688 |
| Vanillin | – | – | 13.022 | 13.022 |
| p- Coumaric acid | 2.3465 | – | 14.131 | 14.177 |
| Ferulic acid | 3.649 | – | 15.241 | 15.296 |
| Ellagic | 3.226 | – | 16.329 | 16.588 |
| Benzoic acid | – | – | 17.297 | 17.297 |
| o- Coumaric acid | 3.5171 | 1.47982 | 18.001 | 17.946 |
| Salicylic acid | – | 7.40213 | 19.447 | 19.459 |
| Cinnamic acid | 2.6637 | – | 22.242 | 22.543 |

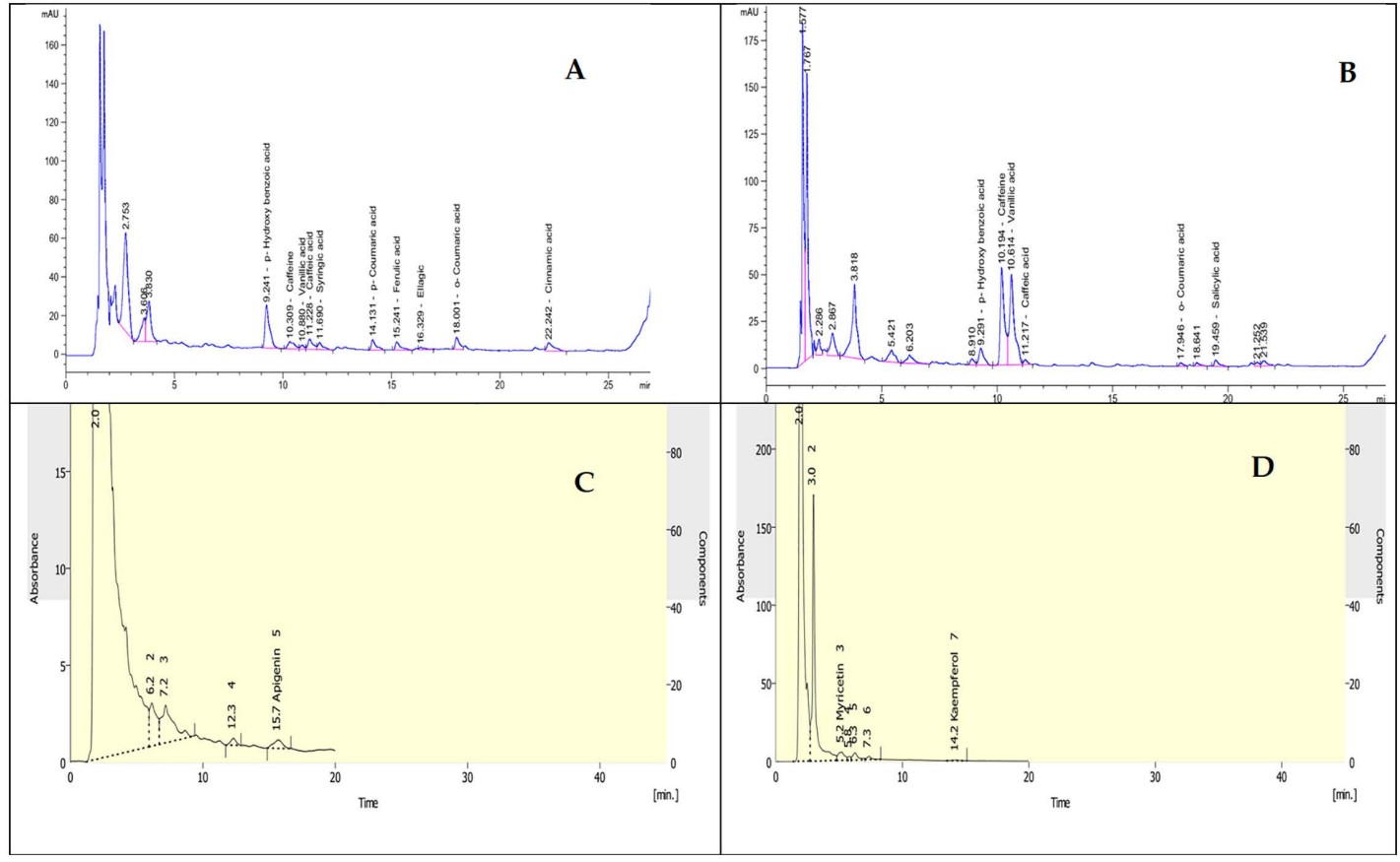

**Fig 1. Bioactive compounds and flavonoids identified in *Arthrospira platensis* and *Chlorella vulgaris*. (A)** Bioactive compounds identified from the *A. platensis*. **(B)** Bioactive compounds identified from the *C. vulgaris*. **(C)** Flavonoids identified from the *C. vulgaris*. **(D)** Flavonoids identified from the *A. platensis*.

nm. The flavonoids contents, functioned at room temperature, (Table 2 and Fig 1C and 1D). The injected volume was 20 µl. Detection: UV detector set at 273 nm action of phenolic compound. *A. platensis* and *C. vulgaris* obtained from phyto-plankton Lab, National Institute of Oceanography (NIOF), Egypt.

## 2.2 Experimental diets preparation

Three diets were formulated. The first diet served as the control (basal diet) and did not include any phenolic compounds. The other two experimental diets were each supplemented with 50 mg/kg of PPA (Diet 1) and 50 mg/kg of PCC (Diet 2), respectively. The chemical composition of the experimental diets was assessed in accordance with [39], and gross energy was assessed in according to [40], as shown in Table 3.

## 2.3 Fish and feeding protocol

Nile tilapia (1.52±0.10 g) were bought and acclimatized for two weeks before being fed commercial diets contained (303.05 g/kg CP and 66.2 g/kg CL) at a rate of 3% of total biomass three times a day at 9:30 a.m., 11:30 a.m., and 3:30 p.m. according to [41] and [42]. The feeding trial was conducted at National Institute of Oceanography and Fisheries (NIOF), Egypt. All the experiments conducted follows NIOF ethical Committee for the care of Aquatic Animals. Following

**Table 2. Flavonoids identified from the *A. platensis* and *C. vulgaris*.**

| Name | Amount (ug ml⁻¹) | | Retention Time (min) | |
|---|---|---|---|---|
| | *A. platensis* | *C. vulgaris* | *A. platensis* | *C. vulgaris* |
| | | – | 1.950 | 2.017 |
| | | – | 2.967 | 6.150 |
| Myricetin | 12.385 | – | 5.200 | 7.183 |
| | | – | 5.817 | 12.317 |
| Apigenin | | 2.341 | 6.250 | 15.683 |
| | | | 7.333 | |
| Kaempferol | 0.926 | | 14.233 | |

**Table 3. Ingredients and proximate composition of the basal diet (g kg diet⁻¹ on dry matter base).**

| Ingredients | Protein in ingredient % | Control | Diet 1 | Diet 1 |
|---|---|---|---|---|
| | | 0 | (50 mg PCA kg diet⁻¹) | (50 mg PCC kg diet⁻¹) |
| Fish meal | 65% | 100 | 100 | 100 |
| Soybean meal | 44% | 370 | 370 | 370 |
| Corn gluten meal | 62% | 60 | 60 | 60 |
| Yellow corn | 8.5% | 250 | 250 | 250 |
| Wheat bran | 14% | 120 | 119.95 | 119.95 |
| Fish oil | | 40 | 40 | 40 |
| Starch | | 30 | 30 | 30 |
| Vitamin and minerals¹ | | 30 | 30 | 30 |
| PCA | | – | 0.05 | – |
| PCC | | | | 0.05 |
| Proximate analysis | | | | |
| Protein | | 303.05 | 303.06 | 303.04 |
| Lipid | | 66.2 | 66.1 | 66.3 |
| Ash | | 48.71 | 48.68 | 48.69 |
| Fiber | | 45.15 | 45.01 | 44.96 |
| Neutral detergent fiber (NDF) | | 159 | 158 | 158 |
| Acid detergent fiber (ADF) | | 97 | 96 | 97 |
| Nitrogen free extract (NFE) | | 536.89 | 537.15 | 537.01 |
| Gross energy MJ kg⁻¹ | | 18.978 | 18.99 | 18.98 |

¹Vitamin and mineral mixture kg⁻¹ of mixture contains described in [34].

PCA: phenolic compound extracted from *Arthrospira platensis*; PCC: phenolic compound extracted from *Chlorella*.

acclimation, Nile tilapia with an initial body weight of 1.52±0.10 g were randomly distributed into three treatment groups, each with three replicates, for a 70-day feeding trial. Each aquarium was stocked with 12 fish, and approximately 20% of the water was renewed daily. The tested diets were provided for the experimental fish satiation three times daily. At the end of the experiment FI was calculated by determined the amount of feed consumed by each fish and expressed as a total. Water quality parameters were monitored during the feeding experiment. Water temperature (°C) and dissolved oxygen (DO, mgL⁻¹) were measured during the experiment trial using a mercury thermometer suspended at a depth of 15 cm

and dissolved oxygen (Keison Company, UK). The pH was recorded twice daily at 8:00 a.m. and 4:00 p.m. using an Orion pH meter (Abilene, Texas, USA). Weekly, water samples were collected to measure the total ammonia (mg $L^{-1}$) using a DREL, 2000 spectrophotometer (Hash Company, Loveland, CO, USA) according to [43]. Water quality parameters were maintained within the recommended range according to [44].

## 2.4 Growth parameters

After the feeding trial, fish from each tank were collected, counted and initial body weight (g) (IBW) and final body weight (g) (FBW) of individual fish were recorded for all fish/each tank at the start and the end of the experiment; the equations used to calculate these values as follows.

Weight gain (g)WG = final weight (g) – initial weight (g)

Specific growth rate (SGR) = LnW2 – LnW1/t (days), Where, Ln = the natural log; W1 = initial fish weight, W2 = the final fish weight in grams and t = Period in days;

Feed conversion ratio (FCR) = Feed intake (g)/weight gain (g);

Protein efficiency ratio (PER) = Weight gain (g)/protein ingested (g)

Fish survival (%) = 100 (final fish number/initial fish number).

## 2.5 Digestive enzymes activity

After anesthesia with 3-aminobenzoic acid ethyl ester (MS-222; 100 mg/L; Sigma, St. Louis, MO, USA), four fish from each tank were slaughtered, and intestinal samples were immediately collected. The samples were homogenized in 10 volumes (w/v) of ice-cold physiological saline solution and centrifuged at 5,000 × g for 15 min at 4 °C. The resulting supernatant was stored for the analysis of endogenous enzyme activity [45]. Chymotrypsin activity was estimated by using the method of [46] with N-benzoyl-Ltyrosine ethyl ester (BTEE) as substrate at 254 nm. 0.2 ml diluted sample solution was added to 6 ml of 0.0005 M BTEE in Tris buffer (10.55 g $CaCl_2$. $2H_2O$ dissolved in 250 ml 0.2 M Tris [hydroxymethyl] aminomethane, adjusted to pH 7.8, diluted to 1 L, and 432 ml methanol). Also, trypsin activity was measured by using methods of [46] with Na-p-toluenesulfonyl-L-arginine methyl ester (TAME) as substrate at 247 nm. 0.2 ml diluted sample solution was added to 6 ml of 0.00104 M TAME in Tris buffer (1.47 g $CaCl_2$. $2H_2O$ dissolved in 200 ml 0.2 M Tris [hydroxymethyl] aminomethane diluted to 1 L, pH 8.1). Lipase activity was determined as described by [47], and titration method was detailed by using olive oil-gum. Amylase activity was estimated according to [48] at 540 nm, and starch was used as the substrate. One ml of diluted sample was incubated for 3 min with 1% starch (1 g soluble starch and 0.035 g NaCl in 100 ml 0.02 M $Na_3PO_4$, pH 6.9). After 3 min, the reaction was stopped by the addition of 2 ml 3,5-dinitrosalicylic acid reagent. The solution was then heated for 5 min in boiling water and then cooled with 20 ml distilled water added.

## 2.6 Serum biochemical analysis and non-specific activities

After fish were anesthetized, blood samples were collected from the caudal vein of five fish per replicate tank. The samples were allowed to clot overnight at 4 °C and then centrifuged at 2,500 × g for 25 min. The non-hemolysed serum was collected and stored at −20 °C until use. The measurements of ALT, AST, total protein, and albumin according to the methods described by [49] and [50], and [51]. Subtracting total serum albumin from total serum protein yields total serum globulin, according to [51]. By modifying the turbidimetric method [52,53], lysozyme activity was assessed, leukocyte phagocytic function was calculated according to [54].

## 2.7 Measurements of antioxidant activity

After fish were anesthetized, hepatic samples (livers of three fish per replicate) were removed, weighed, homogenized, and rinsed with ice-cold phosphate buffer (1:10; phosphate buffer: pH 7.4, 0.064 M), after anestheitzing the fish with

3-aminobenzoic acid ethyl ester (MS 222, 100 mg/L, Sigma, St. Louis, MO). Based on the [55] method, the homogenate was centrifuged for 10 min at 4°C and 4000 g, and the supernatant was used to assay the activity of superoxide dismutase (SOD). A modified technique of [56] was used to assess the catalase (CAT) activity. Beefily, the assay incubates enzyme-containing samples with a phosphate buffer containing suitable concentrations of H2O2. After a specified incubation period, the assay introduces a mixture of sulfosalicylic acid (SSA) and ferrous ammonium sulfate (FAS) to stop the enzyme reaction. SSA binds to the ferric ions produced from the interaction of FAS and residual peroxide, creating a maroon-colored ferrisulfosalicylate complex. This complex is then measured using a spectrophotometer at 490–500 nm. The activities of glutathione peroxidase (GPx) were assessed according to [57] and total antioxidant capacity (T-AOC) was estimated according to [58].

## 2.8  Gene expression

After fish were anesthetized, liver samples form three fish for each treatment were removed from all studied treatments as well as control and homogenized by Tissue Lyser LT apparatus (QIAGEN; Cat No./ID: 85,600). Total ribonucleic acid (RNA) was extracted from these tissues using RNeasy® Mini kit (Qiagen, Cat No. 74104), based on the manufacturer's protocol provided in the kit. The reverse transcriptase reaction of RNA was conducted for complementary DNA (cDNA) synthesizing according to the protocol of High Capacity cDNA Reverse Transcription Kit (Thermo Fisher Scientific, Waltham, MA), cDNA was stored at − 80 ◦C for further molecular analyses. Table 4 lists the target gene primers for the interleukin 1 (*il-1β*), interferon gamma (*inf-γ*), and heat shock protein 70 (hsp 70) genes as well as the housekeeping gene 18s rRNA. Changes in target gene expression levels were presented as n-fold changes relative to the corresponding controls. Relative gene expression ratios (RQ) were estimated using the formula: $RQ = 2^{-\Delta\Delta CT}$ [59].

## 2.9  Statistical analysis of data

The data were arc-sin-transformed prior to analysis [60–61]; however, data are presented untransformed to facilitate the comparisons. All the data were analyzed using the SAS ANOVA procedure (SAS, version 6.03, Soft Inc., Tusla, OK, USA, SAS, 1996). A one-way analysis of variance (One-way ANOVA) was used to determine whether there was significant variation among the treatments followed by post hoc Tukey test ($p < 0.05$) [60] to evaluate significant differences between treatment means.

**Table 4. Oligonucleotide name and sequence of qRT-PCR primers used in this experiment.**

| Gene | Primer sequence 5′-3′ | Amplicon Size (bp) | Slope | Efficiency (%) | R² | Accession No. |
|------|----------------------|--------------------|-------|----------------|-----|----------------|
| 18s | F:5'-GGTTGCAAAGCTGAAACTTAAAGG-3' | 85 bp | −3.32 | 100 | 0.998 | AF497908.1 |
| | R:5'TTCCCGTGTTGAGTCAAATTAAGC-3' | 256 bp | −3.40 | 96 | 0.997 | |
| INF-γ | F:5'-AAGAATCGCAGCTCTGCACCAT-3' | 296 bp | −3.25 | 103 | 0.995 | XM_005448319.1 |
| | R:5'-GTGTCGTATTGCTGTGGCTTCC-3 | 107 bp | −3.50 | 93 | 0.999 | |
| IL-1β | F:5'-CAAGGATGACGACAAGCCAACC-3' | 85 bp | −3.32 | 100 | 0.998 | XM_003460625.2 |
| | R:5'-AGCGGACAGACATGAGAGTGC-3' | 256 bp | −3.40 | 96 | 0.997 | |
| HSP70 | F:5'-CATCGCCTACGGTCTGGACAA-3' | 296 bp | −3.25 | 103 | 0.995 | ′ FJ207463.1 |
| | R:5'-TGCCGTCTTCAATGGTCAGGAT-3' | 107 bp | −3.50 | 93 | 0.999 | |

INF-γ = interferon gamma, IL-1β = interleukin 1β, HSP70 = heat shock protein 70, β-actin = internal reference gene (house-keeping gene).

The primers efficiency was evaluated in UCSC In-Silico PCR on https://genome.ucsc.edu/cgi-bin/hgPcr.

## 3. Results

### 3.1 Growth performance and feed utilization efficiency

Table 5 shows the performance and feed efficiency of Nile tilapia fed diet supplemented with phenolic compounds extracted from *A. platensis* (PCA) and *C. vulgaris* (PCC). The inclusion of phenolic compounds in experimental treatments had a significant effect on tilapia growth performance parameters. When compared to the control diet, the inclusion of phenolic compounds extracted from *A. platensis* and *C. vulgaris* significantly improved ($P<0.05$) weight gain (WG), final body weight (FBW), specific growth rate (SGR), protein efficiency ratio (PER), feed conversion ratio (FCR) and survival rate (SR). The best values of FBW, WG, SGR, FCR and SR were recorded to fish fed diet supplemented with phenolic compounds extracted from PCA followed by and PCC diets.

### 3.2 Intestinal digestive enzymes activities

The intestinal digestive enzyme activities differed significantly ($P<0.05$) after feeding the fish with the diet supplemented with PCA, in comparison to those fed the PCC-supplemented diet. The highest digestive enzymes activities were observed in fish fed with phenolic extract of PCA ($P<0.05$; Table 6).

### 3.3 Serum biochemical parameters

The dietary addition of PCA or PCC significantly decreased ($P<0.05$) the activities of ALT and AST compared to the control diet (Table 7). However, the serum total protein (TP), albumin (ALB), and globulin (GLOB) contents were significantly higher in fish fed the diet enriched with PCA than in those fed the PCC-enriched diet.

**Table 5. Growth performance and feed utilization of Nile tilapia, *O. niloticus* fed diet supplemented with phenolic components from the *A. platensis* (PCA) and *C. vulgaris* (PCC). Compared to control.**

| Parameters | Control | PCA | PCC | *P*-value |
|---|---|---|---|---|
| Initial body weight g fish$^{-1}$ | 1.53±0.11 | 1.52±0.12 | 1.53±0.12 | 0.589 |
| Final body weight g fish$^{-1}$ | 13.420±0.69[c] | 18.06±0.99[a] | 16.34±0.76[b] | 0.023 |
| Weight gain g fish$^{-1}$ | 11.89±0.36[c] | 16.54±0.35[a] | 14.81±0.36[b] | 0.002 |
| Specific growth rate (% day$^{-1}$) | 3.10±0.01[c] | 3.53±0.02[a] | 3.38±0.02[b] | 0.001 |
| Feed intake g fish$^{-1}$ | 20.41±1.17 | 20.65±1.18 | 20.23±1.12 | 0.091 |
| Feed conversion ratio | 1.72±0.05[a] | 1.25±0.03[c] | 1.37±0.02[b] | 0.003 |
| Protein efficiency ratio | 1.82±0.33[c] | 2.50±0.52[a] | 2.28±0.53[b] | 0.001 |
| Fish survival % | 96.00±0.34[b] | 99.20±0.36[a] | 98.50±0.56[a] | 0.001 |

Means followed by different superscripts in the same row are significantly different ($P<0.05$).

PCA: phenolic compound extracted from *Arthrospira platensis*; PCC: phenolic compound extracted from *Chlorella vulgaris.*

**Table 6. Intestinal digestive enzymes (U/g tissue) of Nile tilapia, *O. niloticus* fed diet supplemented with phenolic components from the *A. platensis* (PCA) and *C. vulgaris* (PCC). Compared to control.**

| Parameters | Control | PCA | PCC | *P*-value |
|---|---|---|---|---|
| Chymotrypsin | 7.13±1.14[c] | 10.96±1.14[a] | 9.23±1.13[b] | 0.001 |
| Trypsin | 33.45±2.36[c] | 42.3±2.36[a] | 39.69±2.19[b] | 0.002 |
| Lipase | 986.20±5.21[c] | 1120.00±4.25[a] | 1109.00±4.23[b] | 0.032 |
| Amylase | 692.08±3.09[c] | 765.90±4.02[a] | 732.30±4.01[b] | 0.021 |

Means followed by different superscripts in the same row are significantly different ($P<0.05$). PCA: phenolic compound extracted from *Arthrospira platensis*; PCC: phenolic compound extracted from *Chlorella vulgaris.*

**Table 7. Serum biochemical parameters of Nile tilapia, *O. niloticus* fed diet supplemented with phenolic components from the *A. platensis* (PCA) and *C. vulgaris* (PCC). Compared to control.**

| Parameters | Control | PCA | PCC | *P*-value |
|---|---|---|---|---|
| ALT† (UL⁻¹) | $53.55 \pm 0.32^a$ | $43.15 \pm 0.21^b$ | $43.50 \pm 0.28^b$ | 0.0017 |
| AST‡ (UL⁻¹) | $13.80 \pm 1.14^a$ | $11.47 \pm 1.12^b$ | $12.52 \pm 1.32^b$ | 0.0134 |
| Total protein (g L⁻¹) | $2.82 \pm 0.04^b$ | $3.54 \pm 0.07^a$ | $3.31 \pm 0.09^{ab}$ | 0.0125 |
| Albumin (g L⁻¹) | $1.54 \pm 0.02^b$ | $1.89 \pm 0.02^a$ | $1.84 \pm 0.05^{ab}$ | 0.0212 |
| Globulin (g L⁻¹) | $1.29 \pm 0.02^c$ | $1.66 \pm 0.02^a$ | $1.47 \pm 0.01^b$ | 0.0003 |

Means followed by different superscripts in the same row are significantly different ($P<0.05$). PCA: phenolic compound extracted from *Arthrospira platensis*; PCC: phenolic compound extracted from *Chlorella vulgaris*. ALT†: alanine aminotransferase; AST‡: aspartate aminotransferase.

### 3.4 Immune parameters responses

The phagocytic, lysozyme and immunoglobin M (IgM) activities were significantly improved in fish fed diet supplemented with phenolic extract of *A. platensis* than other treatments. The highest phagocytic activity, lysozyme and IgM activities were recorded for fish fed with PCA ($P<0.05$; Table 8).

### 3.5 Hepatic antioxidant activities

Compared to the control diet, fish fed diets supplemented with PCA and PCC had significantly higher ($P<0.05$) in antioxidant enzyme activities (Table 9). The highest superoxide dismutase (SOD), catalase (CAT), and total antioxidant capacity (T-AOC) activities were observed in fish fed diet supplemented with phenolic extract of *A. platensis*. However, the glutathione peroxidase (Gpx) activity in fish diets supplemented with PCA or PCC was significantly higher ($P<0.05$) than the control, with no significant difference between PCA or PCC.

### 3.6 Gene expression

The *HSP-70*, *interleukin-1β*, and *interferon gamma* genes expression affected by PCA and PCC are shown in Fig 2. Compared to control, fish fed a diet containing phenolic extract of *A. platensis* and *C. vulgaris* showed significantly ($P<0.05$) up-regulated transcripts of *interferon gamma* and *interleukin 1β*, but *HSP-70* genes were down-regulated. Fish fed phenolic extract of *A. platensis* showed the highest levels of *interferon gamma* and *interleukin 1β* gene expression.

## 4. Discussion

### 4.1 Growth and feed efficiency

The findings of the present trial confirmed that phenolic compounds obtained from *A. platensis* or *C. vulgaris* enhanced fish growth, protein utilization, and survival rate (SR) of Nile tilapia. The present results are consistent with [21] who found that Common carp (*Cyprinus carpio* L.) juveniles fed diets containing polyphenols displayed similar

**Table 8. Immune parameters response of Nile tilapia, *O. niloticus,* fed diet supplemented with phenolic components from the *A. platensis* (PCA) and *C. vulgaris* (PCC). Compared to control.**

| Parameters | Control | PCA | PCC | *P*-value |
|---|---|---|---|---|
| Phagocytes % | $24.333 \pm 0.23^c$ | $40.33 \pm 0.35^a$ | $38.33 \pm 0.66^b$ | 0.001 |
| Lysozyme (U ml⁻¹) | $132.00 \pm 1.13^c$ | $258.33 \pm 1.25^a$ | $235.33 \pm 1.55^b$ | 0.041 |
| IgM (μ ml⁻¹)* | $77.00 \pm 2.58^b$ | $140.667 \pm 2.58^a$ | $138.00 \pm 2.02^{ab}$ | 0.002 |

Means followed by different superscripts in the same row are significantly different ($P<0.05$).

IgM*, Immunoglobulin M.

**Table 9. Hepatic antioxidant activities (Ug$^{-1}$ protein) of Nile tilapia, _O. niloticus_ fed diet supplemented with phenolic components from the _A. platensis_ (PCA) and _C. vulgaris_ (PCC). Compared to control.**

| Parameters | Control | PCA | PCC | P-value |
|---|---|---|---|---|
| SOD¶ | 84.867 ± 1.14c | 134.667 ± 1.23a | 129.33 ± 1.12b | 0.002 |
| CAT* | 125.00 ± 1.98c | 188.00 ± 2.09a | 173.33 ± 2.14b | 0.011 |
| Gpx‡ | 560.00 ± 6.21b | 590.02 ± 6.47a | 583.00 ± 6.45a | 0.001 |
| T-AOC† | 17.00 ± 1.45c | 29.44 ± 1.42a | 24.33 ± 1.27b | 0.002 |

Means followed by different superscripts in the same row are significantly different ($P<0.05$). PCA: phenolic compound extracted from _Arthrospira platensis_; PCC: phenolic compound extracted from _Chlorella_ vulgaris. SOD¶: superoxide dismutase, CAT*: catalase, T-AOC†: Total antioxidant capacity, Gpx‡: Glutathione peroxidase.

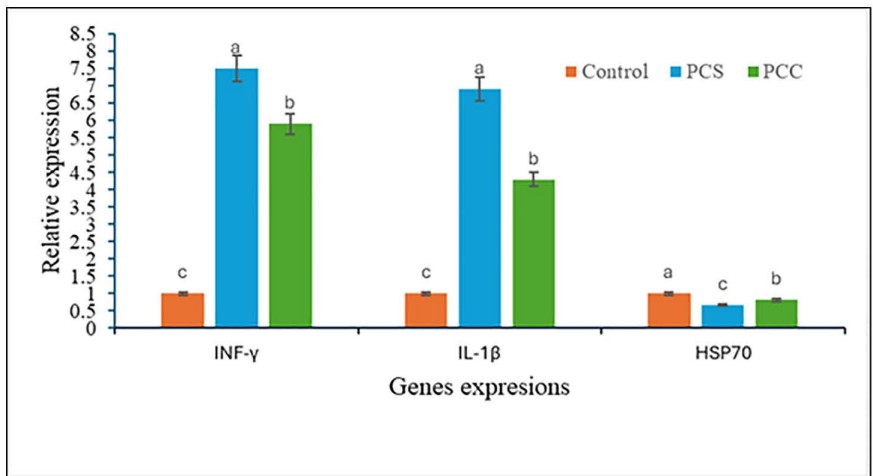

**Fig 2. Relative gene expression of INF-γ, IL-1β and HSP70 of Nile tilapia, _O. niloticus_ fed diets containing different phenolic compounds extracted from _A. platensis_ (PCA) and _C. vulgaris_ (PCC).**

results. Follows the same pattern, [62] stated that Nile tilapia fed diet enriched with chestnut (_Castanea sativa_) polyphenols showed a discernible increase in growth and a decline in FCR. Our findings are in parallels with those of [63], who found that Nile tilapia fed diets supplemented with the polyphenol extracted from grape seed had the lowest FCR (1.31) and improved growth performance. According to the results of the study of [64], the growth performance of common carp in terms of FCR and weight gain was significantly improved at 400 mg kg$^{-1}$ level of polyphenols in canola meal-based diet. Moreover, addition of 0.02% tea polyphenols (TPs) to the diet of coho salmon noticeably boosted their growth performance [35]. [36] found that increasing the amount of _H.fusiformis_ (rich source of polyphenols) in the diet led to better growth in olive flounder. [37] reported higher weight gain in carp was found in diet supplemented with a mixture of Chinese angelica root and astragalus root as sources of polyphenols. [38] found growth performance was significantly higher in Nile tilapia (_O. niloticus_) fed with 1% _Nelumbo nucifera_ (Lotus) peduncle extract (NNPE), rich in polyphenols than control diet. The enhanced growth performance in the present study could be attributed to different scenario as follows: i) the role of polyphenol as an antioxidant agent, ii) polyphenol boost the secretion of mucus that eases the diffusion of nutrients through the absorptive site until arriving the bloodstream [10]., iii) Polyphenols also have an antibacterial effect, which boost the growth of beneficial bacteria and prohibit the growth of pathogens bacteria that assimilate the nutrients through secreted digestive enzymes activity

[65]., iv) Polyphenol improved the role of nutrients absorption participate in different digestive and metabolic pathways required for the vigorous digestive purposes, thus improving growth [66]., v) polyphenolic compounds have a wide range of effects in addition to acting as immune boosters and antioxidants [67,68] and vi) Polyphenols act as growth promoters due to their known effect on humans and their ability to stimulate the metabolism ([9,69].

## 4.2 Intestinal digestive enzymes

Intestinal digestive enzymes have an impact on how fish digest their nutrients [70,71]. The distribution and concentration of digestive enzymes are significantly influenced by intestinal topography and eating patterns [72]. It has been reported that taking herbal supplements rich in polyphenols and natural antioxidants can increase bile secretion, which is essential for proper digestion and absorption, and thus stimulate the activity of digestive enzymes [73]. According to [20], adding medicinal herbs to aquafeed increased amylase activity, resulting in higher carbohydrate digestion and absorption. The current study supports the finding that the increased feed conversion ratio is likely due to the activity of the activated amylase [74,75]. The nutrients digestive enzymes activities were higher in the current study when phenolic compounds from *A. platensis* and *C. vulgaris* were supplemented. The changes in growth and feed intake imply that dietary phenolic compounds may alter the intestinal enzymes' digestive activity, enhancing feed digestibility and nutrient absorption. However, [76] stated that the interaction between polyphenol and amylase alters the enzyme's molecular structure, which causes polyphenols to inhibit amylase and reduce its enzymatic activities.

## 4.3 Serum biochemical

Fish nutritional status and immune response can be accurately predicted by serum biochemical parameters [77,78]. Haematological parameters can be used to gauge the health of fish, especially when they are given functional supplements [79]. Liver damage and dysfunction could be recognized through some enzymes involved in different metabolic pathways as amino acid oxidation, liver gluconeogenesis as ALT and AST [80,81]. In comparison to the control diet, dietary addition of phenolic compounds extracted from *A. platensis* and *C. vulgaris* significantly reduced ALT and AST activities. But fish fed diet supplemented with phenolic compound, the serum TP, ALB, and GLOB contents significantly increase. These findings show that neither of the two bioactive substances caused liver impairments and may even be linked to cytokine production, which protects liver cells. The increase contents of serum albumin and globulin in the present study indicates that their proper transport system is responsible for the fish' good health [79]. The blood's osmotic pressure, metabolism, and the transportation of different metabolites and other organic substances are all highly dependent on the serum albumin concentration [34].

## 4.4 Immune parameters responses

The immune system may be utilized as a primary response to measure a fish's resistance to infections [11,82]. Aquatic organisms' mucosal and humoral immune responses are stimulated by dietary beta-carotene extracted from *A. platensis*, which are known to act as immunomodulators [83]. Lysozyme is a key marker secreted by granulocytes and is crucial for fish's non-specific immunity [84]. The total skin protein reflects the activity of the protein derivatives and enzymes involved in mucosal immunity [85]. Secondary metabolites found in herbal bioactive components have been found to have high anti-infection potency and low toxicity [22,86]. Since they are viewed as an alternative environmentally friendly method and reduce the overuse of antibiotics, phytogenic have been gaining importance in the management of fish disease [87]. In the current study, fish fed supplemented diets performed better in terms of phagocytic, lysozyme, and IgM activities than fish fed the basal diet. The fish fed with phenolic extract of *A. platensis* showed the highest percent of phagocytic and highest activity of lysozyme activity, and IgM. This might be because of the lipophilic properties of polyphenols, which cause dysfunctionality on the pathogenic cell walls. According to [88], these substances have the possible to adjust the

enzymes receptor sites and spots linked to active, catabolism, and active proteins to overcome illnesses and improve aquatic organisms' growth and non-specific immune responses [89].

### 4.5 Hepatic oxidative stress

Both *A. platensis* and *C. vulgaris* have high concentrations of various bioactive materials, including phytopigments, *A. platensis* has a well-established ability to act as an antioxidant and ant-stimulants [90–94]. The highest levels of SOD, CAT, T-AOC, and Gpx activity were found in fish fed diets supplemented with phenolic extract. Furthermore, polyphenols can protected fish from disease and oxidative stress, because the antibacterial, antioxidant, and anti-inflammatory properties of polyphenols [95,96]. Antioxidants scavenge reactive free radicals and safeguard living cells from oxidative damage [97] and consequently enhance fish performance.

### 4.6 Gene expression

According to the current study, fish fed a diet containing PCA had significantly up-regulated transcripts for *interferon gamma* and *interleukin 1β*, but their expression of the *hsp-70* gene was down-regulated. Fish fed PCA followed by PCC showed the highest levels of interferon gamma and interleukin 1β gene expression compared with control group. It has been demonstrated that the cyclooxygenase type 2 inhibitor pigment C-phycocyanin regulates the immune system to shield the organism from disease [98]. A powerful pro-inflammatory cytokine known as *il-1β* is a crucial mediator of the inflammatory response in both acute tissue injury and chronic disease [99]. *inf-* causes a range of physiologically important reactions that support immunity [100]. The bioactive compounds extracted from *A. platensis* or *C. vulgaris* may have an immune-protective effect because it increased the beneficial intestinal microflora, which produced antimicrobial peptides and reduced inflammation [34]. Due to the polyphenolic effect, *C. carpio* taken orally with *A. platensis* displayed increased phagocytic activity, superoxide anion production, and the expression of *il-1β* responses [101]. However, the immune protective effect of bioactive compound extracted from *Spirulina* such as β-carotene and phycocyanin may be associated with the increase in the beneficial intestinal biota, which produced the antimicrobial peptides and modulated the inflammatory reaction [34]. The expression of tumor necrosis factor-α (*tnf-α*) gene was elevated in rainbow trout (*Oncorhynchus mykiss*) fed diet supplemented with *S. platensis* [102]. The present study is in line with up-regulation of *tnf-α* gene in carp, *C. carpio* [103] and Nile tilapia [104] in response to *S. platensis*. In relation to immune system activity, apoptosis, and various aspects of the inflammatory response, *hsps*, biomolecular biomarkers, play a larger role in the host response to environmental toxins, food toxins, and specific and non-specific immune responses to bacterial and viral infections. But as of now, there is scant information available regarding the precise impact of algal phenolic compounds on *hsp-70* in fish culture. The results of the current study corroborate the bioactive compound extracted from algae's role in modulating *hps-70* expression in tilapia by showing that fish fed PCA and PCC had significantly downregulated the *hsp-70* expression. A plant-based polyphenolic compound has been shown to increase the production of *hsp-70* and shield the freshwater prawn *M. rosenbergii* and brine shrimp *A. franciscana* from bacterial infection [105]. The extracted bioactive compounds from spirulina and chlorella (β-carotene and phycocyanin) significantly modulating *hsp70* expressions in Nile tilapia [34].

As well as *Chlorella vulgaris and Spirulina platensis could improve the immune response of* Pacific white shrimp Pacific white shrimp [106–107]. A transcription of *hsp-70* mRNA of *Penaeus monodon* fed diet supplemented with β-carotene was significantly higher than control group under hypoxia condition [108]. Rats have shown a consistent novel response to *A. platensis* extract that is like the stress response in terms of *hsp-70* expression [98]. Additional research on the evolving functional genomics approaches is crucial because it will give researchers the tools, they need to develop a complete understanding of how organisms respond to environmental change [109].

## 5. Conclusion

Phenolic compounds (Polyphenols and Flavonoids) derived from PCA or PCC constitute a good source of antioxidant that has a potential value as functional feed additives for tilapia. Thus, either PCA or PCC modulated growth performance, immune responses, and immune gene expression of tilapia, with PCA providing the most effective responses. Therefore, from a practical viewpoint, incorporating algal phenolic extracts into tilapia diets can reduce reliance on synthetic additives or antibiotics, offering farmers a more sustainable and cost-effective strategy for improving fish performance and disease resilience. Furthermore, microalgae are renewable, eco-friendly, and scalable, making them suitable for field-level adoption in commercial aquaculture. Future studies are needed to explore how extracted polyphenols from *Spirulina* and *Chlorella* affect fishes' health; particularly their roles in managing oxidative stress, supporting immune function¸ gene expression, and improving gut health. Research should also focus on finding the optimal dosage levels and understanding their long-term impact on fish performance and well-being.

## Acknowledgments

The authors would like to thank NIOF institute and Benha University for their cooperation during this research.

**Institutional Review Board Statement:** This study was reviewed and approved by the authority of NIOF Committee for Institutional Care of Aquatic Organisms and Experimental Animals (NIOF-AQ4-F-23-P-023). The study was carried out at National Institute of Fisheries and Oceanography, Egypt.

## Author contributions

**Conceptualization:** Eman Y. Mohammady.

**Data curation:** Eman Y. Mohammady.

**Formal analysis:** Mohamed R. Soaudy.

**Investigation:** Mohamed R. Soaudy.

**Methodology:** Eman Y. Mohammady.

**Project administration:** Mohamed S. Hassaan.

**Software:** Mohamed R. Soaudy.

**Supervision:** Mohamed S. Hassaan.

**Visualization:** Eman Y. Mohammady.

**Writing – original draft:** Mohamed A. Elashry, Abeer M.A. Mahmoud, Soaad A. Sabae, Anisa Mitra, Ehab R. El-Haroun.

**Writing – review & editing:** Mohamed S. Hassaan.

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
