## [Decision Letter · Decision Letter 0]

9 Nov 2025

Dear Dr. Hassaan,

Thank you for submitting your manuscript to PLOS ONE. After careful consideration, we feel that it has merit but does not fully meet PLOS ONE’s publication criteria as it currently stands. Therefore, we invite you to submit a revised version of the manuscript that addresses the points raised during the review process.

We look forward to receiving your revised manuscript.

Kind regards,

Mohammed Fouad El Basuini, Professor

Academic Editor

PLOS ONE

Journal Requirements:

Reviewers' comments:

Reviewer's Responses to Questions

**Comments to the Author**

1. Is the manuscript technically sound, and do the data support the conclusions?

Reviewer #1: Yes

Reviewer #2: Yes

Reviewer #3: Yes

2. Has the statistical analysis been performed appropriately and rigorously?

Reviewer #1: Yes

Reviewer #2: Yes

Reviewer #3: Yes

3. Have the authors made all data underlying the findings in their manuscript fully available?

Reviewer #1: Yes

Reviewer #2: Yes

Reviewer #3: Yes

4. Is the manuscript presented in an intelligible fashion and written in standard English?

Reviewer #1: Yes

Reviewer #2: Yes

Reviewer #3: No

Reviewer #1: Title: should be short; the current title is too lengthy.

The introduction needs mandatory expansion (due to limited background explained) with recent works.

Please add one additional row in table no. 3 for the % of ingredients added, not write with the ingredients.

Tables no. 5, 6, 7, 8, and 9: Footnotes should be added for all of the major headings, which were already used in the table header.

Why did the authors analyze limited genes (table 4)?

Why do authors make italics from page nos. 78-75? If that has an error, make it correct.

Methodologies of the targeted parameters are not just completed from writing estimated according to xxx et al., 2000 (whatever they wrote). It is necessary to write the full methods, whichever authors used. Why are authors reading another article for your data analysis methodology? And also, why are they giving you a citation when you're whitening like this? So I suggest to the authors to write full details of each and every parameter in the methodology section.

P letter in P value should be capitalized and italic along the whole manuscript.

Figures Nos. 5.6 and 7 need to be merged into one graph for better understanding for the readers. Also, they require the footnotes for the analysis parameters and also for the markers names.

Interpretation needs to improve in the section “4.6. Gene expression” with recent research work published.

All table titles should be in professional styles, not like any sentence; look at table no. 1 title.

Gene parameters should be italic along the whole manuscript.

Most of the references of the reference list are without DOI no., why? Even DOIs are available for the concerned articles that are already listed as without DOI reference in the list.

The conversation is incomplete. Section 4.5, Hepatic Oxidative Stress, ends abruptly with an unfinished sentence. This section is crucial for describing the mechanism underlying the reported antioxidant results (PCA group had the highest SOD, CAT, and T-AOC activity). The authors must include the entire material in this part to adequately substantiate their conclusions.

Revise the acronyms used for extracts across the document.

Section 2.3 of the Materials and Methods contains a slight inconsistency in the initial fish weight. The text first cites an average weight of "1.32 g", but later specifies the experimental beginning weight as "(1.52±0.10 g)", which fits with the Abstract. Please utilize the same, most exact starting weight for the trial throughout the text.

Change "A few data were found in this text, which showed the comparative effects between phenolic compounds from Arthrospira platensis and Chlorella vulgaris on aquatic animals". to "Limited data are available on the comparative effects of phenolic compounds from Arthrospira platensis and Chlorella vulgaris on aquatic animals."

Please ensure that the link between the down-regulation of HSP-70 genes in the Results and the concept of reduced stress is clearly explained in the gene expression section. The authors simply indicate that HSP-70 genes were down-regulated, which is a significant discovery that should be interpreted as evidence of improved welfare/lower physiological stress.

Please provide the ethical approval number.

The authors can benefit from these recent studies:

https://doi.org/10.1007/s10499-023-01298-y

https://doi.org/10.1016/j.aqrep.2024.102606

Reviewer #2: I have completed the requested review of the manuscript “Phenolic compounds from Arthrospira platensis and Chlorella vulgaris enhance growth, digestive function, antioxidant capacity, and immune-related gene expression in Nile Tilapia” (PONE-D-25-50895). This study used extracts from the two microalgae named in the title as feed supplements and examined a range of physiological responses after 70 days.

Overall, the study design and methodology are appropriate and the results are clear. However, there are some issues with the presentation of the results and statistics, as well as numerous typographical and mechanical writing errors throughout the manuscript. I’ve tried to highlight those that impact understanding by the reader below, but the authors need to go over the entire document and revise it as needed to remove the many careless mistakes and inconsistencies.

Below I provide specific feedback for the authors which I hope will help improve the manuscript, focusing only on scientific content and writing errors which create confusion about the study. In the absence of line numbers, I have organized my comments by section.

Abstract:

“PCA and” is erroneously italicized in multiple places

Introduction: Reference [1] does not appear to provide evidentiary support for the preceding statement.

References 20, 23, and 24 at the end of the first paragraph are not about polyphenols, and it is not clear why they are included in this list. Given the large number of references included in the bibliography of this standard-length manuscript, I recommend the authors review the cited works with a narrower focus on their specific logical arguments, and curate the bibliography accordingly.

The majority of the second paragraph, starting with “Recently, some research in aquaculture. . .” until the very last sentence, would make more sense being incorporated along with the information on polyphenols at the end of the first paragraph of the Introduction. I suggest combining and streamlining this information to improve the flow of the Introduction overall.

I’m not sure what is meant by “A few data were found. . .”?

The importance of the primary research question of this study is not clear from the information given in the Introduction. Why is this work being done in tilapia specifically? Why were the metrics that are reported selected (what aspects of physiology are relevant and why)? Why were the two microalgae chosen specifically?

Section 2.1:

Please provide complete details on the HPLC protocol. What standard was used? What flow rate? What carrying solvent? Also the phrasing “The flavonoids contents, functioned at 35 C” is very confusing and needs to be rewritten.

Section 2.5:

Remove “for slaughtered fish”

Remove “Hummel (1959)” since reference is already given as [46]

Section 2.6:

Remove redundant phrase at end of first sentence “after anaesthetizing. . .”

Please describe the modifications to [56]

Section 3.1:

Be specific in phrasing to make sure your meaning accurately reflects your findings. “The phenolic extract from A. platensis” was not “found to have the highest SGR”, the fish that were provided diets supplemented with PCA were.

Section 3.2:

Remove Table 6 caption

Section 3.4:

The result presented in Table 8 is the percentage of phagocytes. This is not the same thing as activity, so the related parts of the text (here and also section 4.4) should be revised as necessary.

Section 4.1:

This section is difficult to discern the point the authors are trying to make, apart from the fact that their overall growth results are similar to previous work in other fish provided polyphenol-supplemented diets. If indeed this is the only point, this paragraph can and should be made much more concise.

Also in this paragraph there are several instances of very odd word choice that must be resolved: “Follows the same patterns”, “Indeed, a small addition”, “stashed digestive enzymes activity”, “humanizing growth”.

Several citations at the end of the paragraph are followed by a period and comma in sequence, please correct.

“polyphenolic compounds have a wide range of effects. . .” – such as? Please provide additional details and link them to the results of this study.

Section 4.6:

The authors state that the HSP-70 expression is related to phycocyanin, but this is the first mention of the compound, and is not one of the extracted compounds listed in Table 1.

Remove citation “Ardicli et al., 2022” since it is already given as a bracketed numeral

Conclusion:

I recommend writing out the full name of the abbreviations “PCA or PCC” here, as many readers will read this section in isolation.

Figures 1-4: remove “the” from captions

Figure 5: remove “different” from caption, it creates unnecessary confusion and presumably all treatments were in fact exposed to the same types and proportions of polyphenols

Tables 6-9: header should be PCA, not PCS

Tables 5-9: The way SEM is reported here does not make sense. Is this a combined value for all of the treatments overall (which is meaningless, and inappropriate for metrics in which there is a statistically significant difference)? The SEM for each of the 3 treatments compared by ANOVA should be reported independently with their means to allow the reader to properly comprehend the statistical results.

Reviewer #3: This is an interesting study focused on the use of phenolic compounds as supplements in practical diets for Nile tilapia. The work addresses only the potential benefits of PC supplementation, without exploring different supplementation levels. While this leaves room for future research, it does not add further relevance to the present study. That being said, the study has merit, is interesting, and is well structured. However, the manuscript is not clearly written in several sections, which significantly reduces its clarity and readability. I have provided several comments, but the manuscript would greatly benefit from a thorough revision of the English language.

Abstract:

“or” not italic.

Growth performance, not just performance

Original sentence: “Therefore, three diets were formulated to be isoproteic and isolipidic, and the experimental fish were fed these diets to satiation. The basal diet did not contain any supplements (control diet), while the other experimental diets were supplemented with 50 mg/kg of PCA and PCC, respectively.”

Suggestion: “Therefore, three isoproteic and isolipidic diets were formulated, and the experimental fish were fed these diets to satiation. The basal diet did not contain any supplements (control diet), whereas the other two diets were supplemented with 50 mg/kg of PCA and PCC, respectively.”.

“Compared with control diets…” Just one control diet. Suggestion: “Compared with the control diet, growth parameters and survival rates were significantly enhanced (P < 0.05) by dietary supplementation of phenolic compounds of PCA and PCC.”.

“…significantly higher (P < 0.05)…”

Nile Tilapia not just Tilapia

Suggestion: “Nile tilapia is the second most farmed fish species worldwide.”.

Suggestion: “With the increasing global demand for tilapia [4], farmers have adopted intensive culture techniques for its production. Consequently, the industry has become more susceptible to pathogen infections and disease outbreaks [5].”.

This original sentence is rather confusing: “Among the natural additive phytochemicals, dietary polyphenols and polyphenol-rich diets in have been shown to be promising in modern aquaculture [6, 7] to hunt O2 and N that produced free radicals, modify the activity of antioxidant enzymes and affect cell-to-cell signaling to maintain healthy metabolic functions and improving the fish performance without causing any environmental harm [8-10]. Suggestion: “Among natural phytochemical additives, dietary polyphenols and polyphenol-rich diets have shown great promise in modern aquaculture [6,7]. They can scavenge oxygen and nitrogen-derived free radicals, modulate antioxidant enzyme activity, and influence cell-to-cell signaling, thereby supporting healthy metabolic functions and improving fish performance without causing environmental harm [8–10].”

Suggestion: “Polyphenols, a major class of phytochemicals and secondary metabolites produced by plants and microalgae, are known for their antioxidant and pigmenting properties [13].”

According to the journal guidelines is just: [17-27].

“…C. vulgaris and A. platensis are among…”

“The inclusion of polyphenols in canola meal-based diet at 400 mg kg level improved the growth performance of common carp in terms of FCR and weight gain.” The reference for this sentence is missing.

“when diet was…” or “when diets were…”?

“n Nile tilapia (Oreochromis niloticus)” The species has already been mentioned in the text, so include the scientific name where mentioned for the first time.

2.1. Extraction of phenolic compound. This section describes the detection of the extraction of phenolic compounds in the algae, but not their extraction. There is no mention of solvents, extraction ratios, or procedures (e.g., methanol, ethanol, acetone extraction, sonication, centrifugation, or filtration), which are essential components of an extraction method.

Original sentence: “Three diets were formulated, the first diet (basal diet) was without inclusion of PPA and PPC. The other experimental diets each received a supplement of 50 mg kg diet PPA (Diet 1) and 50 mg kg diet PCC (Diet 2), respectively.”.

In the abstract the authors refer to the first diet as the control diet. Obviously that the control diet is a basal diet without supplementation, but to improve clarity that need to be stated. Suggestion: Three diets were formulated. The first diet served as the control (basal diet) and did not include any phenolic compounds. The other two experimental diets were each supplemented with 50 mg/kg of PPA (Diet 1) and 50 mg/kg of PCC (Diet 2), respectively.

Original sentence: “Nile tilapia (1.32 g)...” All fish had 1.32g? No variation at all? No s.d.?

Original sentence: “Nile tilapia (1.32 g) were bought and acclimatized for two weeks before being fed commercial diets contained (303.05 g/kg CP and 66.2 g/kg CL) at a rate of 3% of total biomass three times a day at 9:30 a.m., 11:30 a.m., and 3:30 p.m. for two weeks according to [41] and [42].” Double mention to the two week period, please revise to improve clarity.

Suggestion: “Following acclimation, Nile tilapia with an initial body weight of 1.52 ± 0.10 g were randomly distributed into three treatment groups, each with three replicates, for a 70-day feeding trial. Each aquarium was stocked with 12 fish, and approximately 20% of the water was renewed daily.”.

Original sentences: “The tested diets were provided for the experimental fish satiation three times daily. The amount of feed consumed by each fish over the feed intake was calculated and expressed as a total.” Confusing description, please revise.

Original sentences: “At the beginning and conclusion of the trial, growth parameter and feed utilization values were recorded; the equations used to calculate these values are shown in the footnote of Table 5.”. This statement is somewhat unclear. Do you mean that the fish were sampled at the start and at the end of the experiment? That can only be inferred, but it is not explicitly stated. In addition, it is not mentioned whether the fish were fed at 3% of body weight per day, as described for the acclimation period. If so, this represents a specific daily feeding rate that should have been recorded to allow accurate calculation of feed utilization parameters. All of these details need to be clearly described in the methodology.

2.4. Growth parameters – In this study, the analysis of growth parameters appears to have received less attention from the authors compared with the other sections. However, the growth performance results are among the most important outcomes of the trial, and the supporting analyses should serve to corroborate these findings. Therefore, providing only a brief description and limited discussion of the growth parameters undermines the overall impact of the study. Greater emphasis and detail should be devoted to this aspect.

Original sentences: “After fish were anesthetized by using 3-aminobenzoic acid ethyl ester (MS 222, 100 mg/L, Sigma, St. Louis, MO), for slaughtered fish, then samples of intestine from four fish in each tank were immediately homogenized in 10 volumes (w/v) of ice-cold physiological saline solution and centrifuged at 5,000 g for 15 min at 4°C; then, the supernatant was stored for endogenous enzymes activity analysis [45].”

“Suggestion: “After anesthesia with 3-aminobenzoic acid ethyl ester (MS-222; 100 mg/L; Sigma, St. Louis, MO, USA), four fish from each tank were slaughtered, and intestinal samples were immediately collected. The samples were homogenized in 10 volumes (w/v) of ice-cold physiological saline solution and centrifuged at 5,000 × g for 15 min at 4 °C. The resulting supernatant was stored for the analysis of endogenous enzyme activity [45].”.

2.6. Serum biochemical analysis and non-specific activities

Suggestion: “After anesthesia (as described above), blood samples were collected from the caudal vein of five fish per replicate tank. The samples were allowed to clot overnight at 4 °C and then centrifuged at 2,500 × g for 25 min.”.

2.7. Gene expression

Please adjust the first sentence, similarly to the suggestion made in the previous comment.

Original sentence: “Table 5 shows the performance and feed efficiency of Tilapia fed phenolic compounds extracted from A. platensis (PCA) and C. vulgaris (PCC)”. Fish were not fed just with phenolic compounds. Therefore, Table 5 shows the performance and feed efficiency of Tilapia fed with the three experimental diets, including the two supplemented with the phenolic compounds. Please adjust the sentence for clarity.

3.1. Growth performance and feed utilization efficiency –

The description of the results related to the growth performance and feed utilization of Nile Tilapia fed the experimental diets is only briefly addressed. The authors provide neither a general overview of the values obtained nor a clear mention of the differences, most of which appear to be statistically significant, as indicated in Table 5, among the three tested diets. They merely state that significant differences occurred, without elaborating further, leaving the reader to infer these details from Table 5.

Regarding FBW, WG, SGR, and PER, the results were significantly different among the three diets, indicating that the inclusion of the phenolic compound PCA led to better outcomes than PCC. The authors note that “the inclusion of phenolic compounds extracted from A. platensis and C. vulgaris improved weight gain (P < 0.05),” which is accurate; however, they fail to mention that differences also existed between the two phenolic compound–supplemented diets. Later, they state that “the phenolic extract from A. platensis was found to have the highest SGR and protein efficiency ratio,” which is again correct, but no reference is made to the statistical significance of these differences compared to the other treatments.

3.2. Intestinal digestive enzymes activities

Original sentence: “The intestinal digestive enzyme activities significantly differed after feeding the fish with phenolic extract from A. platensis followed by C. vulgaris.”.

If the authors have defined the phenolic extracts from A. platensis and C. vulgaris with the acronyms PCA and PCC, these should be used consistently throughout the text to avoid unnecessary repetition. Similarly to one of the comments above, fish were fed experimental diets supplemented with phenolic compounds (PCs), not with PCs alone. Therefore, it would be more accurate to state: “The intestinal digestive enzyme activities differed significantly (P < 0.05) after feeding the fish with the diet supplemented with PCA, in comparison to those fed the PCC-supplemented diet.”.

3.3. Serum biochemical parameters

Suggestion: “The dietary addition of PCA or PCC significantly decreased (P < 0.05) the activities of ALT and AST compared to the control diet (Table 7). However, the serum total protein (TP), albumin (ALB), and globulin (GLOB) contents were significantly higher in fish fed the diet enriched with PCA than in those fed the PCC-enriched diet.”.

3.5. Hepatic antioxidant activities

“Compared to the control diet, fish fed diets supplemented with PCA and PCC had significantly higher…”.

“…significantly higher (P < 0.05)…”.

“Compared to control diet, fish fed a diets containing PCA and PCC showed significantly…”.

Discussion:

“Follows the same patterns,…”. “Following the same pattern…”, maybe?

Hizikia fusiformis. Already mentioned in page 9. Please abbreviate.

The full mention to the common name and scientific name of the Nile tilapia (Oreochromis niloticus) has been done on page 9, please abbreviate. Same with Cyprinus carpio, pages 13 and 16.

4.1. Growth and feed efficiency

Much of the text presented here is a repetition of the introduction. Regardless the fact that it is relevant to include this information to adequately discuss the results from the present study, the repetition should be avoided.

.

Reviewer #1: **Yes:** El-Sayed Hemdan EissaEl-Sayed Hemdan EissaEl-Sayed Hemdan EissaEl-Sayed Hemdan Eissa

Reviewer #2: No

Reviewer #3: No

---

## [Author Response · Author response to Decision Letter 1]

12 Feb 2026

Review Comments to the Author

Reviewer #1:

Comment 1# Title: should be short; the current title is too lengthy.

Response: authors shorted it to be Phenolic compounds from Arthrospira platensis and Chlorella vulgaris enhance growth, digestive function, antioxidant capacity, and immune-related gene expression in Nile Tilapia

Comment2# The introduction needs mandatory expansion (due to limited background explained) with recent works.

Response: authors added recent work about this topic in introduction section see introduction

Comment 3# Please add one additional row in table no. 3 for the % of ingredients added, not write with the ingredients.

Response: Done

Comment 4# Tables no. 5, 6, 7, 8, and 9: Footnotes should be added for all of the major headings, which were already used in the table header.

Response: Done

Comment 5# Why did the authors analyze limited genes (table 4)?

Response: We analyzed the current genes which was related to immune responses of Nile tilapia where we used immune stimulants additive (phenolic compounds) extracted from different algae types. Also the research in the same line measured the same genes see the research according to Hassaan et al. 2021

https://doi.org/10.1016/j.fsi.2020.11.012

Comment 6# Why do authors make italics from page nos. 78-75? If that has an error, make it correct.

Response: We made italic forms for the scientific name of algae only.

Methodologies of the targeted parameters are not just completed from writing estimated according to xxx et al., 2000 (whatever they wrote). It is necessary to write the full methods, whichever authors used. Why are authors reading another article for your data analysis methodology? And also, why are they giving you a citation when you're whitening like this? So I suggest to the authors to write full details of each and every parameter in the methodology section.

Response: Done

Comment 7# P letter in P value should be capitalized and italic along the whole manuscript.

Response: Done

Comment 8# Figures Nos. 5.6 and 7 need to be merged into one graph for better understanding for the readers. Also, they require the footnotes for the analysis parameters and also for the markers names.

Response: author changed it

Comment 9# Interpretation needs to improve in the section “4.6. Gene expression” with recent research work published.

Response: authors improved the section of gene expression in the discussion section

Comment 10# All table titles should be in professional styles, not like any sentence; look at table no. 1 title.

Response: Done

Comment 11# Gene parameters should be italic along the whole manuscript.

Response: Done

Comment 12# Most of the references of the reference list are without DOI no., why? Even DOIs are available for the concerned articles that are already listed as without DOI reference in the list.

Response: Authors checked list references and found two references without DOI because it was international conference and another one is book.

Munglue, P. (2014, November). Effects of dietary Nelumbo nucifera (lotus) peduncle extract on growth performance of Nile tilapia (Oreochromis niloticus). In Proceedings of the 1st Environmental and Natural Resources International Conference (ENRIC 2014), Bangkok, Thailand (pp. 6-7). https://www.researchgate.net/publication/269988078_Effects_of_Dietary_Nelumbo_nucifera_Lotus_Peduncle_Extract_on_Growth_Performance_of_Nile_Tilapia_Oreochromis_niloticus

Boyd, C. E., 1990. Water quality in ponds for aquaculture. Auburn, AL: Alabama Agricultural Experiment Station, Auburn University. https://agris.fao.org/search/en/providers/123819/records/64735ab208fd68d5460241b1

Comment 13# The conversation is incomplete. Section 4.5, Hepatic Oxidative Stress, ends abruptly with an unfinished sentence. This section is crucial for describing the mechanism underlying the reported antioxidant results (PCA group had the highest SOD, CAT, and T-AOC activity). The authors must include the entire material in this part to adequately substantiate their conclusions.

Response: authors added it

Revise the acronyms used for extracts across the document.

Response: Done

Comment 14# Section 2.3 of the Materials and Methods contains a slight inconsistency in the initial fish weight. The text first cites an average weight of "1.32 g", but later specifies the experimental beginning weight as "(1.52±0.10 g)", which fits with the Abstract. Please utilize the same, most exact starting weight for the trial throughout the text.

Response: Done, see line 112

Comment 15# Change "A few data were found in this text, which showed the comparative effects between phenolic compounds from Arthrospira platensis and Chlorella vulgaris on aquatic animals". to "Limited data are available on the comparative effects of phenolic compounds from Arthrospira platensis and Chlorella vulgaris on aquatic animals."

Response: Done, see lines 89-91

Comment 16# Please ensure that the link between the downregulation of HSP-70 genes in the Results and the concept of reduced stress is clearly explained in the gene expression section. The authors simply indicate that HSP-70 genes were down-regulated, which is a significant discovery that should be interpreted as evidence of improved welfare/lower physiological stress.

Response: authors done it

Comment 17# Please provide the ethical approval number.

Response: The ethical approval number already found in line 397 (NIOF-AQ4-F-23-P-023)

The authors can benefit from these recent studies:

https://doi.org/10.1007/s10499-023-01298-y

https://doi.org/10.1016/j.aqrep.2024.102606

Response: author done

Reviewer #2: I have completed the requested review of the manuscript “Phenolic compounds from Arthrospira platensis and Chlorella vulgaris enhance growth, digestive function, antioxidant capacity, and immune-related gene expression in Nile Tilapia” (PONE-D-25-50895). This study used extracts from the two microalgae named in the title as feed supplements and examined a range of physiological responses after 70 days.

Comment 1# Abstract: “PCA and” is erroneously italicized in multiple places

Response: Done

Comment 2# Introduction: Reference [1] does not appear to provide evidentiary support for the preceding statement.

Response: thanks for this comment, authors means chemicals extracted from spirulina such phycocyanin

Comment 3# References 20, 23, and 24 at the end of the first paragraph are not about polyphenols, and it is not clear why they are included in this list. Given the large number of references included in the bibliography of this standard-length manuscript, I recommend the authors review the cited works with a narrower focus on their specific logical arguments, and curate the bibliography accordingly.

Response:

[20] Gian CF, Maliwat SF, Velasquez SMD, Buluran M, Tayamen M, Ragaza JA (2021) Growth and immune response of pond-reared giant freshwater grawn Macrobrachium rosenbergii post larvae fed diets containing Chlorella vulgaris. Aquac Fish 6:465–470. https://doi.org/10.1016/j.aaf.2020.07.002

[23] Aly SM, ElBanna NI, Fathi M (2023) Chlorella in aquaculture: challenges, opportunities, and disease prevention for sustainable development. Aquaculture International pp, 1–28. https://doi.org/10.1007/s10499-023-01229-x

[24] Eissa, E. S. H., Khattab, M. S., Elbahnaswy, S., Elshopakey, G. E., Alamoudi, M. O., Aljàrari, R. M., ... & Naiel, M. A. (2024). The effects of dietary Spirulina platensis or curcumin nanoparticles on performance, body chemical composition, blood biochemical, digestive enzyme, antioxidant and immune activities of Oreochromis niloticus fingerlings. BMC Veterinary Research, 20(1), 215.‏

Comment 4# The majority of the second paragraph, starting with “Recently, some research in aquaculture. . .” until the very last sentence, would make more sense being incorporated along with the information on polyphenols at the end of the first paragraph of the Introduction. I suggest combining and streamlining this information to improve the flow of the Introduction overall.

Response: authors done it

Comment 5# I’m not sure what is meant by “A few data were found. . .”?

Response: We changed to limited data; we meant that the research on this point is limited to obtain the important of the present study.

Comment 6# The importance of the primary research question of this study is not clear from the information given in the Introduction. Why is this work being done in tilapia specifically?

Response: authors thanks reviewer for this comment, especially Nile tilapia are the most widely cultivated of any farmed fish

Comment 7# Why were the metrics that are reported selected (what aspects of physiology are relevant and why)? Why were the two microalgae chosen specifically?

Response: author was chosen this the parameters which were reflects the growth, nutrient utilization as well as health statues of fish while author choice these microalgae because limited data are available.

Comment 8# Please provide complete details on the HPLC protocol. What standard was used? What flow rate? What carrying solvent? Also the phrasing “The flavonoids contents, functioned at 35 C” is very confusing and needs to be rewritten.

Response: authors added it in M&M and corrected see M&M.

Comment 9# Remove “for slaughtered fish”

Response: Done

Comment 10# Remove “Hummel (1959)” since reference is already given as [46]

Response: Done, the reference deleted

Comment 11# Remove redundant phrase at end of first sentence “after anaesthetizing. . .”

Response: Done

Comment 11# Please describe the modifications to [56]

Response: author added that.

Comment 12# Be specific in phrasing to make sure your meaning accurately reflects your findings. “The phenolic extract from A. platensis” was not “found to have the highest SGR”, the fish that were provided diets supplemented with PCA were.

Response: Done, see lines 198-206

Comment 13# Remove Table 6 caption

Response: Done

Comment 14# The result presented in Table 8 is the percentage of phagocytes. This is not the same thing as activity, so the related parts of the text (here and also section 4.4) should be revised as necessary.

Response: Done, see line 315-316

Comment 15# This section is difficult to discern the point the authors are trying to make, apart from the fact that their overall growth results are similar to previous work in other fish provided polyphenol-supplemented diets. If indeed this is the only point, this paragraph can and should be made much more concise.

Response: authors shorted it

Comment 16# Also in this paragraph there are several instances of very odd word choice that must be resolved: “Follows the same patterns”, “Indeed, a small addition”, “stashed digestive enzymes activity”, “humanizing growth”.

Response: The words changed in the text of manuscript.

Several citations at the end of the paragraph are followed by a period and comma in sequence, please correct.

Response: Done

The authors state that the HSP-70 expression is related to phycocyanin, but this is the first mention of the compound and is not one of the extracted compounds listed in Table 1.

Response: authors corrected it.

Remove citation “Ardicli et al., 2022” since it is already given as a bracketed numeral

Response: The reference deleted

Comment 17# I recommend writing out the full name of the abbreviations “PCA or PCC” here, as many readers will read this section in isolation.

Response: Done ,the abbreviation replaced by full name

Figures 1-4: remove “the” from captions

Response: Done

Comment 18# Figure 5: remove “different” from caption, it creates unnecessary confusion and presumably all treatments were in fact exposed to the same types and proportions of polyphenols

Response: Done

Comment 19# Tables 6-9: header should be PCA, not PCS

Response: Done

Tables 5-9: The way SEM is reported here does not make sense. Is this a combined value for all of the treatments overall (which is meaningless, and inappropriate for metrics in which there is a statistically significant difference)? The SEM for each of the 3 treatments compared by ANOVA should be reported independently with their means to allow the reader to properly comprehend the statistical results.

Response: Done

Reviewer #3:

Abstract:

Comment 1# “or” not italic.

Response: Done, see line 19

Comment 2# Growth performance, not just performance

Response: Done, see line 20

Comment 3# Original sentence: “Therefore, three diets were formulated to be isoproteic and isolipidic, and the experimental fish were fed these diets to satiation. The basal diet did not contain any supplements (control diet), while the other experimental diets were supplemented with 50 mg/kg of PCA and PCC, respectively.”

Suggestion: “Therefore, three isoproteic and isolipidic diets were formulated, and the experimental fish were fed these diets to satiation. The basal diet did not contain any supplements (control diet), whereas the other two diets were supplemented with 50 mg/kg of PCA and PCC, respectively.”.

Response: Done, see line 22-25

Comment 4# “Compared with control diets…” Just one control diet. Suggestion: “Compared with the control diet, growth parameters and survival rates were significantly enhanced (P < 0.05) by dietary supplementation of phenolic compounds of PCA and PCC.”.

Response: Done, see line 25-27.

Comment 5# “…significantly higher (P < 0.05)…”

Response: Done, see line 35

Comment 6# Nile Tilapia not just Tilapia

Response: Done, see line 50 and 54

Comment 7# Suggestion: “Nile tilapia is the second most farmed fish species worldwide.”.

Response: Done, see line 50-51

Comment 8# Suggestion: “With the increasing global demand for tilapia [4], farmers have adopted intensive culture techniques for its production. Consequently, the industry has become more susceptible to pathogen infections and disease outbreaks [5].”.

Response: Done, see line 53-55

Comment 9# This original sentence is rather confusing: “Among the natural additive phytochemicals, dietary polyphenols and polyphenol-rich diets in have been shown to be promising in modern aquaculture [6, 7] to hunt O2 and N that produced free radicals, modify the activity of antioxidant enzymes and affect cell-to-cell signaling to maintain healthy metabolic functions and improving the fish performance without causing any environmental harm [8-10]. Suggestion: “Among natural phytochemical additives, dietary polyphenols and polyphenol-rich diets have shown great promise in modern aquaculture [6,7]. They can scavenge oxygen and nitrogen-derived free radicals, modulate antioxidant enzyme activity, and influence cell-to-cell signaling, thereby supporting healthy metabolic functions and improving fish performance without causing environmental harm [8–10].”

Response: Done, see line 55-60

Comment 10# Suggestion: “Polyphenols, a major class of phytochemicals and secondary metabolites produced by plants and microalgae, are known for their antioxidant and pigmenting properties [13].”

Response: Done, see line 62-63

Comment 11# According to the journal guidelines is just: [17-27].

Response: Done, see line 70

Comment 12# “…C. vulgaris and A. platensis are among…”

Response: done, see line 75

Comment 13# “The inclusion of polyphenols in canola meal-based diet at 400 mg kg level improved the growth performance of common carp in terms of FCR and weight gain.” The reference for this sentence is missing.

Response: Done

Comment 14# “when diet was…” or “when diets were…”?

Response: Done, see line 86

Comment 15# “n Nile tilapia (Oreochromis niloticus)” The species has already been mentioned in the text, so include the scientific name where mentioned for the first time.

Response: Done, see line 88

Comment 16# 2.1. Extraction of phenolic compound. This section describes the detection of the extraction of phenolic compounds in the algae, but not their extraction. There is no mention of solvents, extraction ratios, or procedures (e.g., methanol, ethanol, acetone extraction, sonication, centrifugation, or filtration), which are essential components of an extraction method.

Response: the methods was according to Li et al. (2007).

Comment 17# Original sentence: “Three diets were formulated, the first diet (basal diet) was without inclusion of PPA and PPC. The oth

---

## [Decision Letter · Decision Letter 1]

5 Apr 2026

Phenolic compounds from Arthrospira platensis and Chlorella vulgaris enhance growth, digestive function, antioxidant capacity, and immune-related gene expression in Nile Tilapia

PONE-D-25-50895R1

Dear Dr. Hassaan,

We’re pleased to inform you that your manuscript has been judged scientifically suitable for publication and will be formally accepted for publication once it meets all outstanding technical requirements.

Kind regards,

Mohammed Fouad El Basuini, Professor

Academic Editor

PLOS One

Additional Editor Comments (optional):

Reviewers' comments:

Reviewer's Responses to Questions

**Comments to the Author**

Reviewer #3: All comments have been addressed

Reviewer #4: All comments have been addressed

2. Is the manuscript technically sound, and do the data support the conclusions?

Reviewer #3: Yes

Reviewer #4: Yes

3. Has the statistical analysis been performed appropriately and rigorously?

Reviewer #3: Yes

Reviewer #4: Yes

4. Have the authors made all data underlying the findings in their manuscript fully available?

Reviewer #3: Yes

Reviewer #4: Yes

5. Is the manuscript presented in an intelligible fashion and written in standard English?

Reviewer #3: Yes

Reviewer #4: Yes

Reviewer #3: Minor comments:

Line 312-315 – It should be: “Compared with the control diet, dietary supplementation with phenolic compounds extracted from A. platensis and C. vulgaris significantly reduced ALT and AST activities. However, fish fed diets supplemented with these phenolic compounds showed significantly increased serum TP, ALB, and GLOB levels.”

Line 344 - anti-stimulants

Line 346-347 – it should be: “Furthermore, polyphenols can protect fish from disease and oxidative stress, because…”.

Reviewer #4: All the comments and suggestions have been taken into account and the revised manuscript is appropriate and shows scientific merit.

.

Reviewer #3: No

Reviewer #4: No

---

## [Editor Report · Acceptance letter]

PONE-D-25-50895R1

PLOS One

Dear Dr. Hassaan,

I'm pleased to inform you that your manuscript has been deemed suitable for publication in PLOS One. Congratulations! Your manuscript is now being handed over to our production team.

Kind regards,

on behalf of

Prof. Mohammed Fouad El Basuini

Academic Editor

PLOS One